# Differential Urinary Microbiome and Its Metabolic Footprint in Bladder Cancer Patients Following BCG Treatment

**DOI:** 10.3390/ijms252011157

**Published:** 2024-10-17

**Authors:** Kyungchan Min, Chuang-Ming Zheng, Sujeong Kim, Hyun Kim, Minji Lee, Xuan-Mei Piao, Young Joon Byun, Yunjae Kim, Yanghyun Joo, Beomki Cho, Seongmin Moon, Won Tae Kim, Ho Won Kang, Hansoo Park, Seok Joong Yun

**Affiliations:** 1Department of Biomedical Science and Engineering, Gwangju Institute of Science and Technology, Gwangju 61005, Republic of Korea; minchance@gm.gist.ac.kr (K.M.); sujeong1996@gm.gist.ac.kr (S.K.); hkim0719@gm.gist.ac.kr (H.K.); dldlfvnd0614@gist.ac.kr (M.L.); kyj1113@gm.gist.ac.kr (Y.K.); wndidgus@gm.gist.ac.kr (Y.J.); chobk1015@gm.gist.ac.kr (B.C.); 2Department of Urology, Chungbuk National University College of Medicine, Cheongju 28644, Republic of Korea; zcm941311@naver.com (C.-M.Z.); phm1013@hotmail.com (X.-M.P.); lenic0819@naver.com (Y.J.B.); wtkimuro@chungbuk.ac.kr (W.T.K.); howon98@naver.com (H.W.K.); 3Department of Urology, Chungbuk National University Hospital, Cheongju 28644, Republic of Korea; msn0502@naver.com; 4Department of Convergence of Medical Science, Chungbuk National University College of Medicine, Cheongju 28644, Republic of Korea; 5Genome and Company, Seongnam 13486, Republic of Korea

**Keywords:** Urinary microbiome, 16S rRNA gene sequencing, Bladder cancer, Intravesical BCG treatment, Quinolone, *Bifidobacterium*

## Abstract

Recent studies have identified a urinary microbiome, dispelling the myth of urine sterility. Intravesical bacillus Calmette–Guérin (BCG) therapy is the preferred treatment for intermediate to high-risk non-muscle-invasive bladder cancer (BCa), although resistance occurs in 30–50% of cases. Progression to muscle-invasive cancer necessitates radical cystectomy. Our research uses 16S rRNA gene sequencing to investigate how the urinary microbiome influences BCa and its response to BCG therapy. Urine samples were collected via urethral catheterization from patients with benign conditions and non-muscle-invasive BCa, all of whom underwent BCG therapy. We utilized 16S rRNA gene sequencing to analyze the bacterial profiles and metabolic pathways in these samples. These pathways were validated using a real metabolite dataset, and we developed predictive models for malignancy and BCG response. In this study, 87 patients participated, including 29 with benign diseases and 58 with BCa. We noted distinct bacterial compositions between benign and malignant samples, indicating the potential role of the toluene degradation pathway in mitigating BCa development. Responders to BCG had differing microbial compositions and higher quinolone synthesis than non-responders, with two *Bifidobacterium* species being prevalent among responders, associated with prolonged recurrence-free survival. Additionally, we developed highly accurate predictive models for malignancy and BCG response. Our study delved into the mechanisms behind malignancy and BCG responses by focusing on the urinary microbiome and metabolic pathways. We pinpointed specific beneficial microbes and developed clinical models to predict malignancy and BCG therapy outcomes. These models can track recurrence and facilitate early predictions of treatment responses.

## 1. Introduction

Bladder cancer (BCa) ranks among the top ten most commonly diagnosed cancers worldwide, with an estimated 550,000 new cases annually [1]. Key risk factors include advanced age, male gender, cigarette smoking, and exposure to occupational carcinogens, like benzene and toluene [2,3]. BCa is categorized into non-muscle-invasive (NMIBC) and muscle-invasive (MIBC) subtypes based on tumor invasion into the bladder wall, and further stratified by tumor–node–metastasis (TNM) stage and grade (low and high) [3]. NMIBC, though less aggressive than MIBC, often recurs after treatment and can progress to muscle-invasive status, necessitating radical cystectomy [4]. The five-year recurrence rates for NMIBC range from 50% to 70%, with progression rates between 10% to 30% [5].

The standard treatment for NMIBC includes surgical tumor resection followed by intravesical bacillus Calmette–Guérin (BCG) therapy, derived from a live attenuated *Mycobacterium bovis* [4]. NMIBC can be categorized into low-, intermediate-, and high-risk groups based on pathological stage, grade, tumor size and number, presence of concomitant carcinoma in situ (CIS), lymphovascular invasion, and variant histological features according to the American Urological Association risk stratification [6]. For patients with intermediate- and high-risk NMIBC, the administration of intravesical BCG is highly recommended [4]. While there is no universally established regimen, the Southwest Oncology Group (SWOG) regimen for intravesical BCG treatment consists of a 6-week induction course followed by maintenance therapy. The maintenance involves three weekly instillations at 3, 6, 12, 18, 24, 30, and 36 months [4,7]. However, 30–50% of patients do not respond to BCG therapy [8], and for those failing BCG treatment, radical cystectomy is often the next step, significantly impacting quality of life [4]. Given the inconsistent supply of BCG and the absence of comparable alternatives [9], there is a growing research focus on finding substitutes or methods to overcome BCG resistance.

The human microbiome, a complex community of diverse microorganisms, has become an important point of research due to its role in numerous physiological processes and its association with multifactorial diseases [10]. Emerging evidence suggests that dysbiosis, or microbial imbalance, plays a role in carcinogenesis [10]. The potential of certain microbiomes to provide anti-cancer benefits has also been highlighted, with microbe-–immune interactions and gut-derived metabolites, like short-chain bile acids, enhancing antitumor immunity and potentially inhibiting cancer progression [11,12]. Representatively, lactic acid bacteria (LAB), comprising genera, like *Lactococcus*, *Lactobacillus*, and *Bifidobacterium*, are renowned for their health benefits. These bacteria produce lactic acid, a metabolite known for its advantages to human health [13].

Contrary to the historical view of a sterile urine, advanced techniques, such as 16S rRNA gene sequencing and expanded quantitative urine culture, have revealed a distinct urinary microbiome [14,15]. This suggests that, as discovered in other cancers, the urinary microbiome could influence the etiology of BCa or the response to treatment. By altering the bacterial composition, either by targeting deleterious species or introducing beneficial ones, therapeutic advantages might be achieved. Our study explores the interaction between BCa, BCG therapy, and the urinary microbiome, including its microbial metabolic pathways. We also concentrated on pinpointing microbes within the urinary microbiome that may be either detrimental or advantageous in the development of BCa or the efficacy of BCG treatment.

## 2. Results

### 2.1. Patient Characteristics Description

The study involved 87 participants, including 29 male patients with BPH and 58 BCa patients. The benign group consisted exclusively of male patients. In the BCa cohort, 29 patients provided paired pre- and post-BCG samples, resulting in 29 matched sets. The other 29 BCa patients contributed unpaired post-BCG samples. Table 1 details the clinical characteristics of these patients. The average age in the benign group was 73.0 years, which was not significantly different from the BCa group’s average age of 72.6 years (*p* = 0.6373). However, there was a notable difference in gender composition, with the BCa group comprising 79.3% male patients (*p* = 0.0070). In terms of antibiotic usage within a month prior to sample collection, 20.7% of the benign group and 10.3% of the BCa group reported usage, with no significant difference observed (*p* = 0.6373). The prevalence of smoking was identical in both groups at 48.3%, showing no statistical disparity (*p* > 0.9999). In addition, none of the patients in either group had a urinary tract infection within a month of sample collection.

Within the BCa group, clinical follow-up averaged 668 days, with a high-grade disease prevalence of 91.4%. The response rate to intravesical BCG treatment was 67.2%, and the recurrence rate stood at 29.3%. The median RFS time was 509.5 days. Additionally, the progression rate was 5.2%, with a median PFS time of 650 days.

### 2.2. Compared to the Benign Group, the BCa Group Has a Lower Toluene Degradation Capacity

According to the Chao1 index, there was no significant difference in microbial richness between the benign and pre-BCG groups; however, a significant difference was observed between the benign and post-BCG groups (*p* < 0.0001), as well as between the pre-BCG and post-BCG groups (*p* < 0.05; Figure 1B). The Simpson index indicated a statistically significant difference only between the benign and pre-BCG groups (*p* < 0.05; Figure 1B). Beta diversity analysis based on Bray–Curtis dissimilarity also revealed a considerable overlap in microbial distribution between the benign and pre-BCG groups, distinct from the post-BCG group (*p* = 0.001, R^2^ = 0.159, Figure 1C). Upon synthesizing the analyses on diversity, it was evident that the benign group and the pre-BCG group displayed some differences yet shared similarities. This distinction was further validated at the species level, as depicted in Figure 1D. The benign group showed a more consistent species-level composition within its samples compared to the greater dissimilarity observed among BCa samples (Figure 1D). For a detailed comparison between groups, the pre-BCG group was considered as the BCa group. This decision was guided by the logic that a comparison between the benign group and the BCa group should be made prior to BCG treatment to exclude the treatment’s impact. The heatmaps reveal divergent microbial composition patterns between the groups at both genus and species levels, as illustrated in Figure 1E and Appendix A. Specifically, the genera *Anaeroplasma*, *Tetragenococcus*, *Thauera*, and *Sporotomaculum* were significantly enriched in the benign group (Figure 1E). Similarly, *Treponema sp5*, *Desulfovibrio mexicanus*, and *Lactobacillus agilis* showed significant enrichment in the benign group (Appendix A).

In addition to microbial composition, our study also explored the enrichment of microbial metabolic pathways within each group. Figure 1F presents a volcano plot that highlights the significant enrichment of specific pathways, including PWY-6145 (the superpathway of CMP-sialic acids biosynthesis), PWY-5789 (3-hydroxypropanoate/4-hydroxybutanate cycle), and PWY-5184 (toluene degradation VI (anaerobic)), in the benign group. Among them, we focused on the toluene degradation pathway because toluene is well-known for its carcinogenic properties [16]. As the genera *Thauera* and *Desulfovibrio mexicanus*, both known for their toluene-degrading capabilities [17], were found to be enriched in the benign group, we deduced that the toluene degradation pathway may play a significant role in preventing bladder carcinogenesis. The toluene degradation pathway is schematically depicted in Appendix A.

To establish the relevance of toluene to bladder carcinogenesis, we utilized an open dataset [18]. From this dataset, we identified metabolites that were significantly enriched in both cancer patients and controls (Appendix A). We then performed ORA using these selected metabolites. The ORA revealed that the metabolites enriched in the BCa group were closely associated with the category of organic acids and derivatives (*p* < 0.001, FDR = 0.012; Appendix A). Metabolites enriched in the control group were closely related to organic acids and derivatives (*p* < 0.001, FDR < 0.001), benzenoids (*p* < 0.001, FDR < 0.001), organoheterocyclic compounds (*p* < 0.001, FDR = 0.002; Appendix A), and exposure to volatile organic compounds (VOCs) (*p* = 0.004, FDR = 0.221; Appendix A). Furthermore, hippuric acid—a degradation product of toluene in the human body [19]—was found at significantly increased levels in urine from benign control patients in this dataset and other studies [20], reinforcing our hypothesis that an impairment in toluene degradation is linked to bladder carcinogenesis.

When comparing male BCa patients with male benign controls, more definitive results were obtained (Appendix A). A significant difference in the Chao1 index was observed between the benign group and both the pre-BCG and post-BCG groups (*p* < 0.05 and *p* < 0.0001, respectively; Appendix A). Additionally, there was a statistically significant difference between the pre-BCG and post-BCG groups (*p* < 0.05, Appendix A). For the Simpson index, a significant difference was noted only between the benign group and the pre-BCG group (*p* < 0.05; Appendix A). Beta diversity also varied between the groups (*p* = 0.001; Appendix A). Differential abundance testing revealed distinct microbial composition patterns between the benign and BCa groups, as illustrated in Appendix A. Microbial metabolic pathways, such as PWY-6145, PWY-5789, and PWY-5184 (toluene degradation VI), were found to be enriched in the benign group, mirroring the patterns observed in the BCa group that included female patients, as shown in Figure 1F (Appendix A).

### 2.3. Responders Exhibit Changes in Urinary Microbiome and Metabolite Production after Intravesical BCG Treatment

To uncover the characteristics of patients who responded to BCG therapy, we conducted comparative analyses between matched pre-BCG and post-BCG samples, with each group comprising 29 patients (Figure 2A). Among these patients, 23 were identified as responders, while 6 were non-responders. Appendix A demonstrate no significant differences in diversities between the paired pre-BCG and post-BCG groups in the ‘Paired cohort’, which includes both responders and non-responders. Figure 2B compares the microbial composition at the species level based on the response to BCG treatment and the timing of sample collection relative to BCG therapy in the paired cohort. Our analysis revealed that while pre-BCG and post-BCG samples from non-responders were almost identical, those from responders displayed significant differences. As such, we proceeded with a more detailed comparison at both the genus and species levels. This revealed variations in microbial composition associated with the response to BCG (Figure 2C,D and Appendix A). In responders, the post-BCG group had more *Klebsiella oxytoca*, *Morganella morganii*, and *Salmonella enterica*, but fewer *Anoxybacillus kestanbolensis* and *Bacillus flexus* (Figure 2D). In the paired cohort, the post-BCG group exhibited an increased presence of *Klebsiella oxytoca*, *Morganella morganii*, *Salmonella enterica*, and *Trabulsiella farmeri* compared to the pre-BCG group (Appendix A). However, non-responders did not exhibit a significant difference in microbial composition between the pre-BCG and post-BCG groups. We concluded that BCG treatment primarily induces compositional changes in the urinary microbiome, observed exclusively in responders and not in non-responders.

Next, we compared metabolic pathways between the groups (Figure 2E and Appendix A). The post-BCG group in responders showed increased activity in pathways PWY-7002 and PWY-4361 (Figure 2E). The S-methyl-5-thio-α-D-ribose 1-phosphate degradation I pathway, also known as PWY-4361, was significantly enriched exclusively in responders (Figure 2E and Appendix A). This pathway degrades S-methyl-5-thio-α-D-ribose 1-phosphate into 2-oxoglutarate and L-methionine, both of which are known for their anticancer properties [21,22,23,24,25,26]. PWY-4361 is schematically depicted in Figure 2F.

Lastly, we identified key driver microbes, revealing significant bacterial shifts from the pre-BCG to post-BCG microbiome. Figure 2G and Appendix A display the common sub-network at the genus level in the responder and paired cohorts, respectively. Enlarged red nodes highlight the important driver microbes that transition from pre-BCG to post-BCG status. *Tindallia_Anoxynatronum*, *Lutispora*, *Marinococcus*, and *Parvimonas* were identified as critical driver microbes in responders with the highest NESH scores, underscoring their significant role in this transformation (Figure 2H).

### 2.4. Responders Exhibit Enriched Quinolone Biosynthesis Post-BCG Treatment

Next, we performed subgroup analyses for both the pre-BCG and post-BCG groups. Initially, we evaluated 29 pre-BCG samples, comprising 23 responders and 6 non-responders, and compared them based on factors, such as gender, grade, and response to BCG (Figure 3A). No significant difference was observed between responders and non-responders in terms of antibiotic use within the month prior to sample collection (*p* > 0.9999) or smoking history (*p* = 0.6693). There were no cases of urinary tract infection within a month before sample collection. While non-responders exhibited a higher age range, this difference was not statistically significant (*p* = 0.0684). However, they did show significantly higher recurrence rates (*p* < 0.0001; Appendix A). Since there were no cases of disease progression in the pre-BCG group, the analysis of recurrence essentially mirrored that of the response to BCG treatment. Alpha and beta diversity assessments revealed no significant differences across these variables. Differential abundance testing, however, identified gender-based variations in microbial composition (Appendix A) and differences associated with recurrence (Appendix A) and BCG response (Figure 3C). Notably, responders exhibited higher abundances of *Campylobacter ureolyticus* and *Bifidobacterium bifidum* species compared to non-responders (Figure 3C; Appendix A). Furthermore, survival analyses focusing on the presence of *C. ureolyticus* and *B. bifidum* revealed no significant extension in RFS (*p* = 0.5536 and 0.2915, respectively; Figure 3E,F). As previously noted, the *Bifidobacterium* genus is renowned for its health benefits, prompting a further examination of its abundance. However, our analysis indicated that the abundance of the *Bifidobacterium* genus did not differ significantly based on treatment response (*p* = 0.8837; Figure 3D). No significant differences were detected in metabolic pathways when categorized by BCG response. In summary, the comparison of pre-BCG samples revealed only minor differences in microbial composition, and no significant changes were observed in metabolic pathways.

The analysis of post-BCG samples included 58 specimens from 39 responders and 19 non-responders (Figure 3A). No significant differences were observed between responders and non-responders in terms of antibiotic use within the month prior to sample collection (*p* = 0.2475), smoking history (*p* = 0.5827), or age (*p* = 0.1832). No cases of urinary tract infection were reported within a month before sample collection. However, non-responders exhibited significantly higher rates of recurrence and progression (*p* < 0.0001 and 0.0314, respectively; Appendix A). We compared these groups based on gender (Appendix A), grade (Appendix A), response (Figure 3B,G–K,M), recurrence (Appendix A), and progression (Appendix A). Although no significant differences were observed in diversities, there were distinct differences in microbial composition and metabolic pathways based on these parameters. The heatmap revealed distinct microbial composition patterns at the family and genus levels according to the response to BCG (Figure 3B). Significantly, responders exhibited a prevalent presence of the *Bifidobacterium* genus, previously mentioned as beneficial bacteria (Appendix A; Figure 3H). Within this genus, various species, including *B. adolescentis*, *B. breve*, and *B. longum*, were notably more prevalent among responders (Appendix A; Figure 3G). Specifically, *B. breve* and *B. longum* were associated with prolonged RFS (*p* = 0.0203 and 0.0423, respectively; Figure 3I,J), but the *Bifidobacterium* genus and *B. adolescentis* were not (*p* = 0.4816 and 0.3012, respectively; Appendix A). Cox regression analysis of post-BCG urine samples identified the presence of *B. breve* as an independent significant factor influencing RFS prolongation (*p* = 0.044; Table 2).

The analysis of metabolic pathway abundances showed an increased presence of PWY-6660 and PWY-6662 pathways in responders, both linked to quinolone biosynthesis (Figure 3K). When comparing enzyme abundance, higher levels of quinolone synthases, such as 2-heptyl-3-hydroxy-4(1H)-quinolone synthase and 2-heptyl-4(1H)-quinolone synthase, were observed in responders compared to non-responders (Figure 3M). These enzymes contribute to the production of 2-heptyl-3-hydroxyl-4-quinolone, a variant of quinolone (Figure 3L). Bold values indicate statistically significant results (*p* < 0.05).

### 2.5. Development of Predictive Models for Malignancy and Response to BCG Treatment

To develop a predictive model for malignancy and response to BCG treatment, we incorporated genera and species identified in differential abundance tests. Initially, we utilized data on the presence and relative abundance of genus *Thauera* and *Desulfovibrio mexicanus*. These microbes, identified to play a key role in the toluene degradation pathway [17], were suggested to be associated with the prevention of bladder carcinogenesis. This model exhibited excellent predictive accuracy for malignancy, achieving an AUC of 0.913 (Appendix A).

We then aimed to develop a model predicting the response to BCG treatment. Initially, we included data on the presence and abundance of *C. ureolyticus* and *B. bifidum* from pre-BCG urine samples (*n* = 29), resulting in a model with good predictive performance (an AUC of 0.717; Appendix A). Subsequently, we incorporated data on the presence and abundance of the *Bifidobacterium* genus and various species, including *B. adolescentis*, *B. breve*, and *B. longum*, from post-BCG urine samples (*n* = 58). This model demonstrated good predictive accuracy for BCG treatment response, as evidenced by an AUC of 0.753 (Appendix A).

## 3. Methods

### 3.1. Sample Collection

The study’s design is illustrated in Figure 1A, following the ethical guidelines of the Declaration of Helsinki. Samples were collected with strict adherence to these principles at Chungbuk National University Hospital (CBNUH; Cheongju, Republic of Korea), after receiving approval from the CBNUH Institutional Review Board (CBNUH 2020-04-011-003). Informed consent was obtained from all participants prior to any invasive procedures. From January 2020 to November 2022, urine samples were prospectively collected from individuals diagnosed with benign prostatic hyperplasia (BPH) or BCa. Eligible participants included those pathologically diagnosed with BPH and NMIBC undergoing intravesical BCG therapy. The exclusion criteria excluded participants with pyuria detected in their urine samples at the time of collection, BCa patients with loss to follow-up, and samples failing to meet the sequencing quality control standards.

Control urine samples were obtained from the surgical suite immediately before the transurethral resection procedures. BCa samples were differentiated as ‘Pre-BCG’ or ‘Post-BCG’, depending on when they were collected, as depicted in Figure 2A. Pre-BCG samples were taken in the surgical suite, after anesthesia and draping, right before starting the transurethral resection of the bladder tumor (TUR-BT). Post-BCG samples were collected via catheterization right before a follow-up cystoscopy, upon the completion of either the induction or maintenance phases of BCG treatment. All samples were collected through sterile urethral catheterization and immediately stored at −24 °C within two hours of collection.

### 3.2. Treatment Protocol for BCa

Patients with intermediate- and high-risk BCa underwent six weekly doses of intravesical BCG therapy as part of the induction process. Cystoscopy was performed three months after completion of BCG induction therapy, and every three months thereafter. If there was no recurrence or progression for two years, it was performed once every six months. If a lesion suggestive of recurrence was detected during cystoscopy, a TUR-BT procedure was performed. In cases where no lesions were detected, patients underwent a maintenance regimen of three weekly BCG treatments at 3, 6, 12, and 18 months following surgery. Computed tomography (CT) images were examined three months after the initial TUR-BT, and subsequent CT examinations were conducted annually. Recurrence-free survival (RFS) and progression-free survival (PFS) were calculated from the time of diagnosis to the first event of recurrence or progression, as determined by clinico-pathological examination.

### 3.3. Defining the BCG Response

BCG responders were defined as patients who did not have any recurrence or disease progression during the treatment and follow-up periods. In this study, disease recurrence was defined as relapse of primary NMIBC of the same pathologic stage, and progression of NMIBC was defined as progression to muscle-invasive disease after disease recurrence. The last follow-up urine sample after BCG treatment was selected for analysis. Conversely, non-responders were defined as patients who had confirmed recurrence or disease progression during any stage of BCG therapy. In non-responders, the urine sample collected at initial diagnosis of recurrence or progression was selected for analysis.

### 3.4. DNA Extraction, Library Construction, and Sequencing

DNA extraction procedures were conducted at CBNUH. Total urinary DNA was isolated utilizing the Quick-DNA™ Urine Kit (Zymo Research, Los Angeles, CA, USA). The sequencing libraries were prepared in alignment with the Illumina 16S Metagenomic Sequencing Library protocols [27], targeting the amplification of the V3 and V4 regions. A comprehensive description of the library preparation methodology is provided in the Appendix A. Following library preparation, sequencing was carried out on an Illumina MiSeq platform (Illumina, San Diego, CA, USA). The sequencing depth for each paired-end FASTQ file ranged from 35 to 100 megabases (Mbp), averaging 65 Mbp.

### 3.5. Bioinformatics Analysis

Raw sequence data underwent quality control and adapter trimming using fastp (Version 0.22.0) [28]. Subsequent to data filtering, alignment to the premade 16S rRNA gene database, which is grounded on the Greengene database, was executed using Kraken2 (Version 2.0.7-beta) [29]. This allowed us to obtain microbial read counts for individual samples. Both alpha and beta diversities were computed and visually represented using the ‘phyloseq’ [30] and ‘microViz’ [31] package in R version 4.2.3 (Vienna, Austria). A computational decontamination process using DNA quantification data was performed using R package ‘decontam’ (Version 3.19) [32]. R package ‘pheatmap’ (Version 1.0.12) [33] and Graphpad Prism version 10.0.1 (Boston, MA, USA) were used to visualize differences in microbial composition between groups. The R package ‘survival’ was employed for survival analysis and Cox regression analysis [34].

To discern ‘driving microbes’ transitioning from pre-BCG samples to their post-BCG counterparts, we employed the Molecular Ecological Network Analysis Pipeline (MENA, http://ieg4.rccc.ou.edu/mena (accessed on 19 October 2023) [35]. This provided insights into the relationships (edges) between constituting microbes (nodes). After refining the edge list, it was fed into NetShift (https://web.rniapps.net/netshift) (accessed on 19 October 2023) [36] to compute the neighbor shift (NESH) score, helping to pinpoint dominant microbes influencing inter-group variations.

For metabolic pathway inference and related enzyme identification, we utilized PICRUSt2 [37]. Illumina adapter sequences of paired-end reads were excised with Cutadapt in QIIME2 [38]. Processed sequences were further handled with quality check and trimming in QIIME2 (Version 2023.2). By leveraging the PICRUSt2 plugin in QIIME2 with default parameters [39], we secured MetaCyc pathways from 16S rRNA communities. Analyses rooted in the KEGG enzyme and MetaCyc databases facilitated the elucidation of differential enzyme activities and metabolic pathways between groups. The R package ‘ggpicrust2’ was used for comparing and visualizing enzyme abundance across various groups [40].

To construct prediction models, we utilized the ‘glm’ function in R for developing logistic regression models. The ‘pROC’ package in R [41] facilitated the computation of the area under the curve (AUC) and the generation of receiver operating characteristic (ROC) curves.

### 3.6. Using an External Dataset

To validate our hypothesis regarding bladder carcinogenesis, we employed an external urine metabolite dataset from the Appendix A of the study conducted by Niziol et al. [18]. We selected metabolites that showed statistically significantly different abundances according to their analysis between urine samples from patients with BCa and those with benign diseases. These selected metabolites were then further analyzed using metabolite set enrichment analysis (MSEA) in MetaboAnalyst 6.0 [42]. The list of metabolites was entered as a one-column dataset for over-representation analysis (ORA).

### 3.7. Statistical Analyses

All statistical evaluations were executed in R and Graphpad Prism. Alpha diversity was assessed via ‘estimate_richness’ in the package ‘phyloseq’ [30], with group comparisons made using the Wilcoxon test. Beta diversity was determined by PERMANOVA based on Bray-Curtis dissimilarity, utilizing ‘adonis2’ in the ‘vegan’ package [43]. Differential abundance analyses were performed using ‘DESeq2’ [44] and ‘EdgeR’ [45]. In our analyses, the Wald test was employed within ‘DESeq2’, while the exact test was executed within ‘EdgeR’. Multiple test corrections were conducted using the Benjamini–Hochberg procedure for both ‘DESeq2’ and ‘EdgeR’. Univariate and multivariate Cox analyses were conducted using the ‘coxph’ function from the ‘Survival’ package in R [34]. An adjusted *p* value below 0.05 was deemed statistically significant. The detailed statistical methodologies employed in this study are thoroughly delineated in the legends accompanying each figure.

## 4. Discussion

In comparing microbial metabolic pathways between the benign and BCa groups, we were unable to discern or infer any relationship between pathways PWY-6145 and 5789 and BCa prevention. However, the enrichment of the toluene degradation VI pathway in the benign group is noteworthy, particularly due to toluene’s established association with BCa [16]. Toluene, a constituent of aromatic hydrocarbons, acts as a precursor to aromatic amines, such as toluidines and diaminotoluenes [46]. A systematic review identifying occupational carcinogens associated with BCa revealed that workers exposed to aromatic amines face significantly higher risks of BCa. Notably, industries with high aromatic amine exposure, like tobacco, dye, and rubber manufacturing, showed elevated BCa risk ranging from 1.49 to 1.72 [17]. Our study further highlights the significance of the toluene degradation VI pathway in the benign group, marked by the presence of *Thauera* and *Desulfovibrio mexicanus*. Both are known for their toluene degradation capabilities [37]. Consequently, the presence of these organisms, along with the enrichment of the toluene degradation pathway, likely contribute to reducing BCa risk while simultaneously elevating the level of hippuric acid (Appendix A). This observation proposes a potential new mechanism for BCa prevention via the urinary microbiome and its metabolic pathways (Figure 4A). The mechanism was validated through the analysis of an external urine metabolite dataset. Metabolites enriched in urine from patients with benign urological diseases and BCa showed a close relationship with VOCs, benzenoids, and organic acid compounds. Given that toluene, a component of VOCs, contains a benzene ring and is degraded into organic acid compounds [47], the correlation established through ORA strongly suggests that the mechanism is closely related to toluene. Additionally, *p*-cresol, which is reported to be elevated in the urine of BCa patients compared to controls [20], is a derivative of toluene modified by a hydroxy group at position 4 [48] and is identified as a potential carcinogen [49]. This supports the link between toluene and bladder carcinogenesis. In summary, we suggested and validated that an increase in toluene-degrading bacteria, such as *Thauera* or *Desulfovibrio* genera in the urinary microbiome, is related to a decreased amount of toluene in urine, leading to a lower risk of bladder carcinogenesis. In addition, based on the mechanism suggested by toluene degradation, we developed a high-performing predictive model for malignancy. This model can be used not only to support the diagnosis of BCa but also to monitor its recurrence.

In our paired cohort, we identified 2-oxoglutarate and L-methionine as being associated with intravesical BCG treatment. 2-oxoglutarate has been recognized for its anticancer properties, primarily through angiogenesis inhibition [21,23]. Moreover, L-methionine has been found to inhibit the growth of various cancer cell lines [22,24,25,26], further underscoring its potential significance in cancer therapy. The inferred enrichment of 2-oxoglutarate and L-methionine in the urine of responders to intravesical BCG treatment could contribute to the treatment’s efficacy, potentially through the anticancer effects of these metabolites. This inference suggests a novel mechanism for BCG response, potentially mediated by alterations in the urinary microbiome and its metabolic pathways.

Several studies on BCa have highlighted the *Bifidobacterium* genus as beneficial [50,51,52]. Notably, James et al. [52] employed a methodology similar to ours, collecting serial catheterized urine samples from 29 NMIBC patients to assess the efficacy of intravesical therapies, including BCG and gemcitabine. Their observation of reduced microbial richness post-treatment, along with elevated *Bifidobacterium* levels in patients without recurrence, aligns with our findings. Our research builds on theirs by offering more comprehensive analyses at the species level. Specifically, we identified certain bacterial species, such as *B. breve* and *B. longum*, that significantly influence RFS. These species may be suitable candidates for intravesical injection of live bacteria, potentially leading to improved treatment outcomes. In another study from our group, we selected a bacterial strain from the *Bifidobacterium* genus and validated its antitumor efficacy through in vitro and in vivo experiments, including subcutaneous and orthotopic BCa mouse models [53]. This study demonstrated elevated expression of pro-inflammatory genes and increased cell populations. Considering that BCG exerts its efficacy via pro-inflammatory genes and trained immunity in the urinary bladder [54], this study supports our findings. Furthermore, we successfully developed prediction models for the response to BCG treatment based on pre-BCG and post-BCG urinary microbial composition data. These models are valuable for assessing the effectiveness of BCG therapy both at the outset and after the completion of either the induction or maintenance phases.

In our study of the post-BCG cohort, we found that the quinolone biosynthesis pathway was more prominent in the urine of those who responded to BCG treatment (Figure 4B). Quinolone antibiotics, known for their high urinary concentration and ability to target type II DNA topoisomerases (including DNA topoisomerase II and IV), have been suggested as effective in treating BCa [55,56]. Additionally, their anti-inflammatory properties may help reduce adverse events in patients undergoing BCG therapy. Supporting this, a randomized study demonstrated that short-term prophylactic use of oral levofloxacin lessened the severity of adverse events and enhanced progression-free and cancer-specific survival in patients receiving intravesical BCG therapy post-TUR-BT [57]. Consequently, the unique microbiome and its metabolic pathways in BCG responders could be indicative of a favorable response to BCG treatment.

However, our study is not without limitations. While the metabolic pathways associated with the BCG response have been supported by multiple references, it is crucial to emphasize that these pathways require validation using empirical metabolomics data. Secondly, unlike previous studies, we successfully identified numerous microbes at the species level using 16S rRNA gene sequencing data. However, it is important to note that this method has limitations in microbial resolution compared to the more advanced shotgun metagenomic sequencing approach. Thirdly, the sample size of our study was relatively small (*n* = 87), and the population was homogenous, consisting solely of Asian individuals. Fourthly, our control group was exclusively composed of male patients diagnosed with BPH. As we gathered samples from urethral catheterization, which is an invasive, possibly painful, and infection-inducing procedure, we were unable to collect samples from healthy controls without urological diseases. In addition, we could not find any appropriate external dataset containing catheterized urine samples from healthy controls. Though it was not a perfect match for controls, BPH was our preferred choice, considering the ease of sample collection, similar age distribution, and male predominance in BCa. To improve the quality of the control group, we excluded patients in the control group with urinary retention, infection, and high prostate-specific antigen (PSA) levels, which suggest possible prostate cancer. Finally, it is important to note that only a small subset of patients in our study experienced progression, with just three cases (*n* = 3) reported.

In summary, our study represents a pioneering effort to unravel the mechanisms underlying the response to intravesical BCG treatment, focusing on the urinary microbiome and its metabolic pathways. Our study suggests that certain *Bifidobacterium* species could be viable candidates for the intravesical injection of live bacteria, which may potentially enhance treatment outcomes. Through the robust predictive model we developed, utilizing urinary microbial composition data, it becomes possible to not only predict malignancy during the initial cystoscopy but also to assess the response to BCG therapy upon completion of either the induction or maintenance phases. Due to the limitations of the study, further research is essential to validate and expand our findings. Future studies should consider employing a shotgun metagenomic approach along with matched metabolomics data. Additionally, incorporating a more diverse participant pool in terms of race and gender, including female subjects, and ensuring a sufficient number of patients with disease progression, will be crucial for achieving a more comprehensive understanding of this subject.

## Figures and Tables

**Figure 1 ijms-25-11157-f001:**
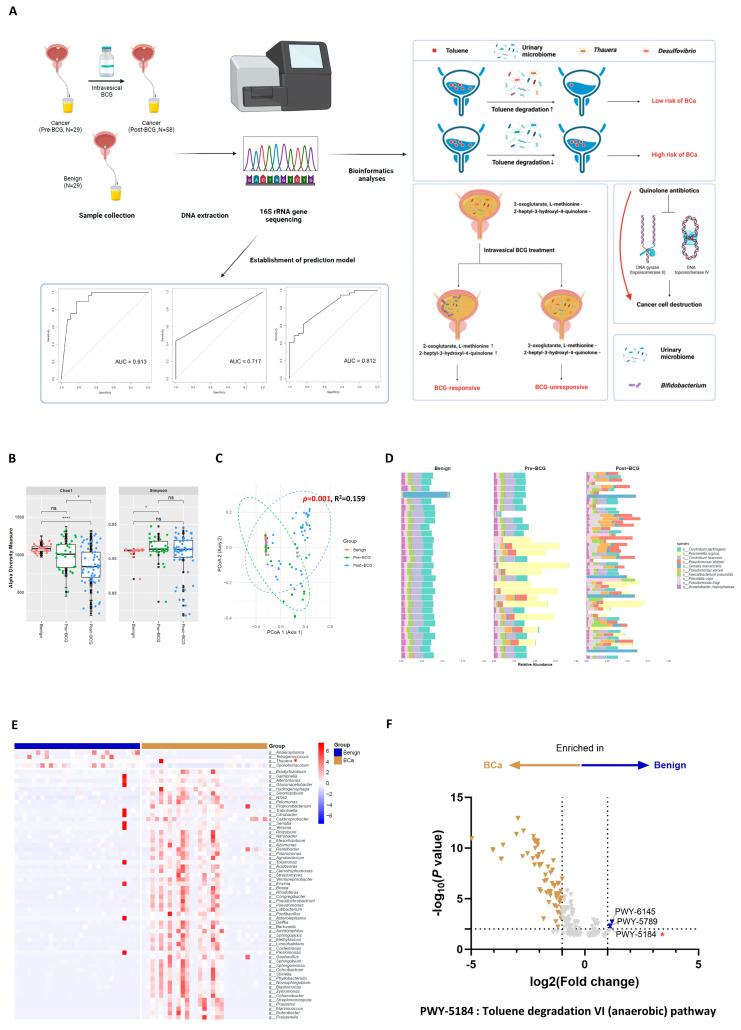
The toluene degradation pathway is enriched in the urinary microbiome of patients without BCa. (**A**) Schematic illustration of the study design. (**B**) Chao1 and Simpson indices of alpha diversity comparison among benign (*n* = 29), pre-BCG (*n* = 29), and post-BCG (*n* = 58) groups. Group differences were analyzed using the Wilcoxon test. Significance levels: * *p* < 0.05, **** *p* < 0.0001. (**C**) Beta diversity analysis between benign (*n* = 29), pre-BCG (*n* = 29), and post-BCG (*n* = 58) groups using Bray–Curtis distance. Beta diversity significance was determined by PERMANOVA with 999 permutations. (**D**) Bar graph depicting microbial composition in each group. The top 10 most abundant species are highlighted and labeled. (**E**) Heatmap displaying microbial composition variations among groups. Colors represent different groups: blue for benign (*n* = 29), and gold for BCa (*n* = 29). Genera selected by DESeq2 with *p* value < 0.05 and |log_2_(Fold change)| > 2 for their differential abundance are annotated. (**F**) Volcano plot of differentially abundant metabolic pathways between benign (*n* = 29) and BCa (*n* = 29) groups. A positive fold change indicates enrichment in the benign group; a negative fold change indicates the opposite. Metabolic pathways selected by EdgeR with −log_10_ (*p* value) > 1 and log_2_ (Fold change) ≥ 2 for their differential abundance are annotated. PWY-6145: superpathway of CMP-sialic acids biosynthesis, PWY-5789: 3-hydroxypropanoate/4-hydroxybutanate cycle, PWY-5184: toluene degradation VI (anaerobic). ns: not significant.

**Figure 2 ijms-25-11157-f002:**
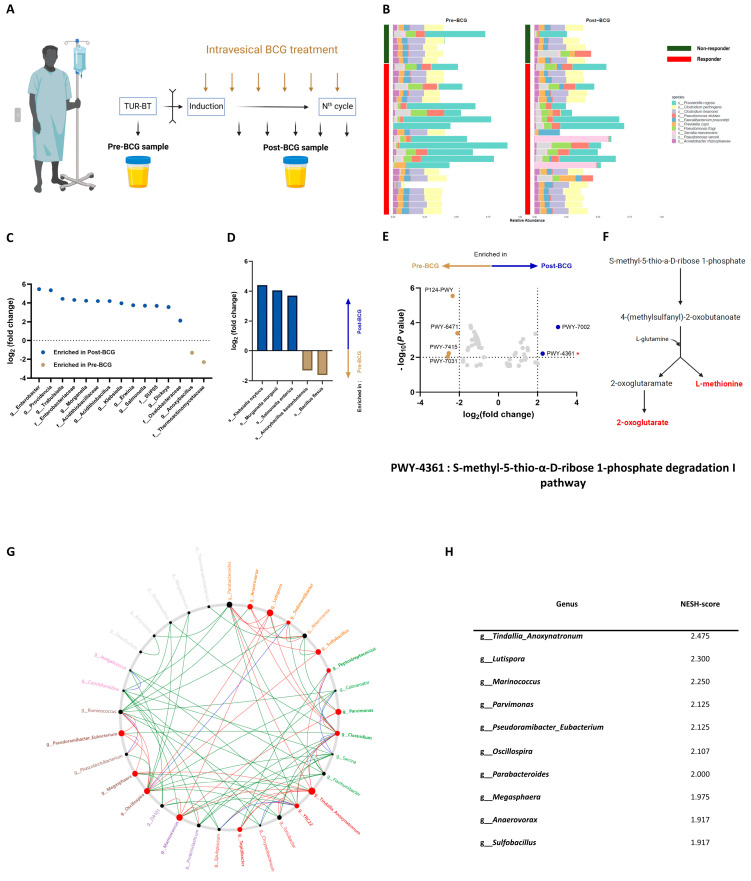
The response to BCG treatment may be associated with L-methionine production by the urinary microbiome. (**A**) Schematic overview of BCa sample collection. Pre-BCG samples were obtained during TUR-BT, and post-BCG samples were gathered during cystoscopy following BCG treatment. (**B**) Composition bar graph displaying the top 10 abundant species in matched samples. The *x*-axis represents the relative abundance of each species, while the *y*-axis categorizes samples based on response. (**C**) Differential microbial abundance at family and genus levels in matched samples among responders (*n* = 23). Microbes enriched in the post-BCG group are highlighted in blue, and those enriched in the pre-BCG group are in gold. Microbes selected by DESeq2 with *p* value < 0.05 and |log_2_ (fold change)| > 2 for their differential abundance are annotated. (**D**) Differential microbial abundance at species level in matched samples among responders (*n* = 23). Microbes enriched in the post-BCG group are highlighted in blue, and those enriched in the pre-BCG group are in gold. Microbes selected by DESeq2 with *p* value < 0.05 and |log_2_ (fold change)| > 2 for their differential abundance are annotated. (**E**) Volcano plot of differentially abundant metabolic pathways between paired pre-BCG (*n* = 23) and post-BCG (*n* = 23) groups. A positive fold change indicates enrichment in the post-BCG group; a negative fold change indicates enrichment in the pre-BCG group. Metabolic pathways selected by EdgeR with −log_10_ (*p* value) > 2 and log_2_ (fold change) > 2 for their differential abundance are annotated. (**F**) Simplified overview of the PWY-4361 pathway (S-methyl-5-thio-α-D-ribose 1-phosphate degradation I pathway). It shows how S-methyl-5-thio-α-D-ribose 1-phosphate is utilized to produce 2-oxoglutarate and L-methionine. (**G**) Identification of potential ‘driver microbes’ influencing BCG treatment response in responders (*n* = 23), derived from bacterial network analysis of the urinary microbiome. This analysis compares pre-BCG and post-BCG groups. Node sizes correspond to scaled NESH scores, with nodes colored red indicating increased betweenness in the transition from pre-BCG to post-BCG microbiomes. Large red nodes highlight key driver taxa behind BCG treatment, with their names bolded. Line colors represent connections, with red edges for associations unique to post-BCG microbiomes, green for those unique to pre-BCG, and blue for associations present in both. (**H**) Top 10 NESH scores of candidate driver microbes at the genus level influencing changes due to BCG treatment.

**Figure 3 ijms-25-11157-f003:**
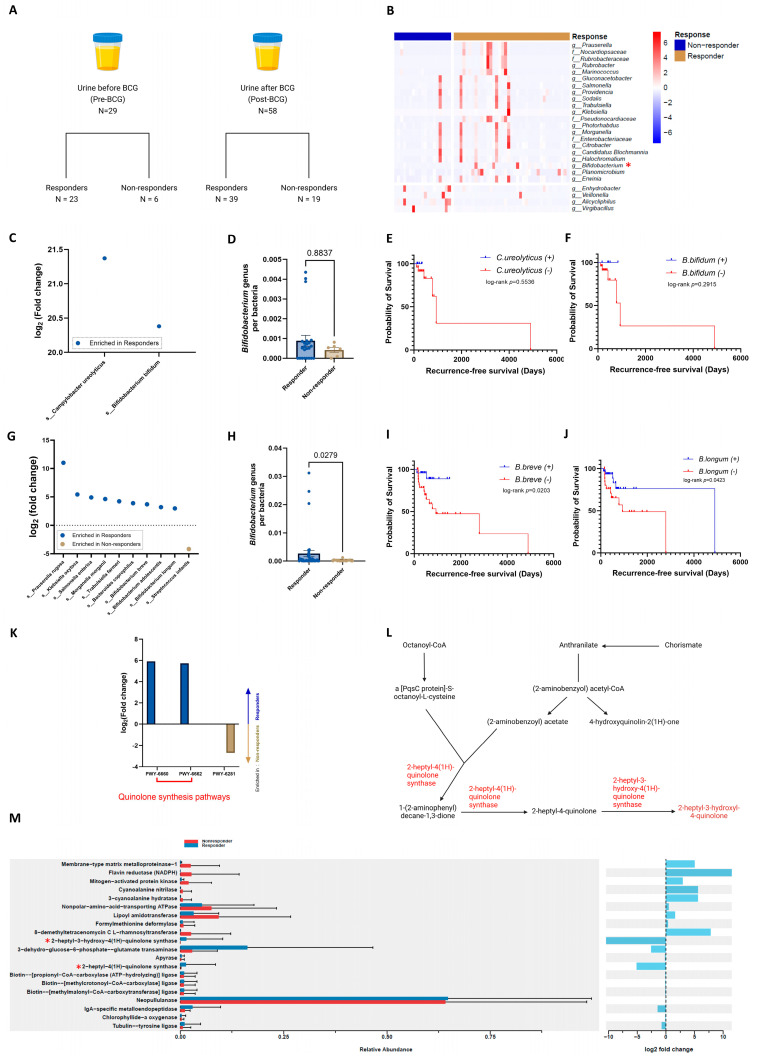
Enrichment of *Bifidobacterium* species and quinolone synthesis pathways in responders is associated with prolonged RFS. (**A**) Representation of urine sample categorization based on the timing of collection and response to BCG treatment. (**B**) Heatmap illustrating variations in microbial composition at the family and genus levels, categorized by response in the post-BCG group (*n* = 58). Colors represent different groups: blue for non-responder, and gold for responder. Microbes selected by DESeq2 with *p* value < 0.05 and |log_2_ (fold change)| > 2 for their differential abundance are annotated. (**C**) Differential microbial abundance between responders (*n* = 23) and non-responders (*n* = 6) in the pre-BCG group. Microbes enriched in responders are highlighted in blue. Microbes selected by DESeq2 with *p* value < 0.05 and |log_2_ (fold change)| > 2 for their differential abundance are annotated. (**D**) Comparison of the relative abundance of the *Bifidobacterium* genus in the pre-BCG group (*n* = 29), analyzed using the Wilcoxon test. (**E**) Kaplan–Meier survival curve depicting outcomes based on the presence of *C. ureolyticus* in the pre-BCG group (*n* = 29), analyzed using the log-rank test. (**F**) Kaplan–Meier survival curve depicting outcomes based on the presence of *B. bifidum* in the pre-BCG group (*n* = 29), analyzed using the log-rank test. (**G**) Differential microbial abundance between responders (*n* = 39) and non-responders (*n* = 19) at the species level in the post-BCG group. Microbes enriched in responders are highlighted in blue and those enriched in non-responders are indicated in gold. Microbes selected by DESeq2 with *p* value < 0.05 and |log_2_ (fold change)| > 2 for their differential abundance are annotated. (**H**) Comparison of the relative abundance of the *Bifidobacterium* genus in the post-BCG group (*n* = 58), analyzed using the Wilcoxon test. (**I**) Kaplan–Meier survival curve depicting outcomes based on the presence of *B. breve* in the post-BCG group (*n* = 58), analyzed using the log-rank test. (**J**) Kaplan–Meier survival curve depicting outcomes based on the presence of *B. longum* in the post-BCG group (*n* = 58), analyzed using the log-rank test. (**K**) Differences in metabolic pathway abundance between responders and non-responders in the post-BCG group (*n* = 58). Pathways that are more abundant in the responders are highlighted in blue, while those with higher abundance in non-responders are indicated in gold. Metabolic pathways selected by EdgeR with −log_10_(*p* value) > 2 and log_2_ (fold change) > 2 for their differential abundance are annotated. (**L**) Schematic diagram of quinolone biosynthesis pathways including PWY-6660 and PWY-6662. Enzymes that are enriched in the responders are colored red above the arrows. (**M**) Enzyme abundance comparison using ggpicrust2 in post-BCG responders (*n* = 39) and non-responders (*n* = 19). This figure illustrates the differences in enzyme abundance between responders and non-responders after BCG treatment, as analyzed with the ggpicrust2 package.

**Figure 4 ijms-25-11157-f004:**
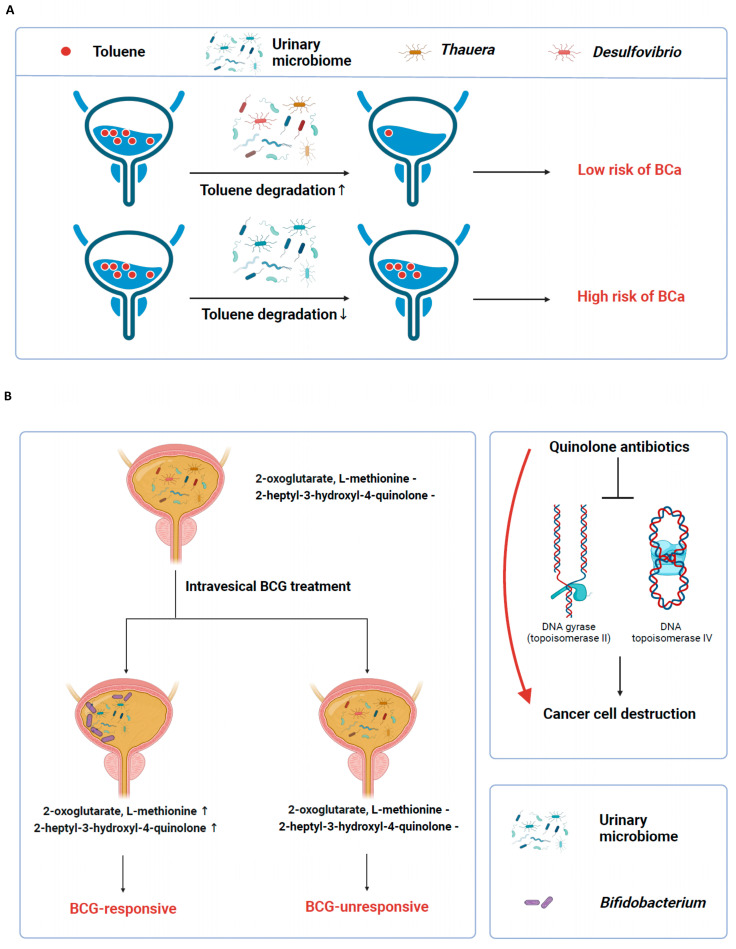
Graphical abstract for the study. This figure serves as a graphical abstract, illustrating the study’s key findings. It shows how the urinary microbiome activates specific metabolic pathways. (**A**) Illustration of the toluene degradation pathway activated by the urinary microbiome of the benign group. This pathway breaks down toluene, a known risk factor for BCa, potentially reducing the incidence of the disease. (**B**) Illustration of increased L-methionine and 2-heptyl-3-hydroxyl-4-quinolone production in responders to intravesical BCG therapy. The figure further highlights the role of quinolone antibiotics in inhibiting type II topoisomerases, contributing to the destruction of cancer cells.

**Table 1 ijms-25-11157-t001:** Clinical characteristics of benign and BCa groups (*n* = 87).

	Benign (*n* = 29)	BCa (*n* = 58)
Age (y), median (IQR) ^a^	73.0 (63.5, 78.5)	73.5 (64.8, 83.0)
Antibiotics usage (1 month) ^b^	6 (20.7%)	6 (10.3%)
Smoking ^c^	14 (48.3%)	28 (48.3%)
Gender, *n* (%) ^d^		
Male	29 (100%)	46 (79.3%)
Female	0	12 (20.7%)
Grade, *n* (%)		
Low		5 (8.6%)
High		53 (91.4%)
T stage, *n* (%)		
Ta		20 (34.5%)
T1		38 (65.5%)
Response to BCG, *n* (%)		
Responder		39 (67.2%)
Non-responder		19 (32.8%)
Recurrence, *n* (%)		
Recurred		17 (29.3%)
Not recurred		41 (70.7%)
Recurrence-free survival (Days), median (IQR)		509.5 (301.8, 777.0)
Progression, *n* (%)		
Progressed		3 (5.2%)
Not progressed		55 (94.8%)
Progression-free survival (Days), median (IQR)		650.0 (394.3, 1101.0)
Follow-up period (Days), median (IQR)		668.0 (411.3, 1232.0)

^a^ The unpaired parametric *t*-test indicates no statistically significant difference between the benign and BCa groups (*p* = 0.6373). ^b^ Analysis using Fisher’s exact test reveals no statistically significant difference between the benign and BCa groups (*p* = 0.2032). ^c^ Analysis using Fisher’s exact test reveals no statistically significant difference between the benign and BCa groups (*p* > 0.9999). ^d^ Fisher’s exact test indicates a significant statistical difference between the benign and BCa groups (*p* = 0.0070).

**Table 2 ijms-25-11157-t002:** Univariate and multivariate Cox proportional hazard models for RFS in BCa patients with baseline samples (*n* = 58).

Variable	Univariate	Multivariate	
	*p* Value	HR (95% CI)	*p* Value
Age (y)	0.448		
Gender	0.783		
Grade	0.616		
T stage	0.095		
Response to BCG	0.998		
Recurrence	0.998		
Progression	0.898		
*Bifidobacterium* presence (post-BCG)	0.484		
*B. adolescentis* presence (post-BCG)	0.307		
*B. bifidum* presence (post-BCG)	0.062		
*B. breve* presence (post-BCG)	**0.036**	**0.214 (0.045, 0.958)**	**0.044**
*B. longum* presence (post-BCG)	0.052		
*Bifidobacterium* abundance (post-BCG)	0.116		
*B. adolescentis* abundance (post-BCG)	0.127		
*B. bifidum* abundance (post-BCG)	**0.045**		0.074
*B. breve* abundance (post-BCG)	0.097		
*B. longum* abundance (post-BCG)	0.094		

## Data Availability

The sequencing data reported in this paper were deposited in the European Nucleotide Archive (accession no. PRJEB72901).

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
