# Peer review of "Differential Urinary Microbiome and Its Metabolic Footprint in Bladder Cancer Patients Following BCG Treatment"

_ijms, 2024, doi:10.3390/ijms252011157_

Round 1
Reviewer 1 Report
Comments and Suggestions for Authors
I have read your manuscript“Differential urinary microbiome and its metabolic footprint in bladder cancer patients following BCG treatment” with great pleasure. This is an interesting article. However, the authors have to address the following before it can be considered.
Major:
1. Are there any comparisons of urine microbiome composition before and after BCG treatment for the same patient in the data? If not, please explain why?
2. Please explain in the manuscript why healthy subjects were not included for comparison in the data.
3. Discussion: Please include a more detailed explanation or possible molecular mechanisms elucidating the proposed relationship between the urinary microbiome, metabolic pathways (including the role of toluene degradation), and BCa.
4.
Minor:
1. Introduction – TNM– Please provide full title before using an abbreviation.
2. I did not see Table 1 in the manuscript, original images, or supplementary file(s).
3. Please remove the yellow highlights from pages 10, 11, and 15.
Author Response
Comments to reviewers
First of all, we would like to thank you for the detailed and thoughtful comments. We discussed the points raised by the reviewers and made some modifications and improvements. All points that we modified or added are highlighted in yellow.
[Major points]
- Are there any comparisons of urine microbiome composition before and after BCG treatment for the same patient in the data? If not, please explain why?
Thank you for the comment. Section 3.3 in the Results section, titled “Responders exhibit changes in urinary microbiome and metabolite production after intravesical BCG treatment,” describes the analysis of urine samples collected from the same patients before and after BCG treatment. The section begins with, “To uncover the characteristics of patients who responded to BCG therapy, we conducted comparative analyses between matched pre-BCG and post-BCG samples, with each group comprising 29 patients.” The results are presented in Figure 2 and Supplementary Figure 3. We observed changes in the microbial composition of responders, but not in non-responders. In terms of metabolites, we found that 2-oxoglutarate and L-methionine were associated with intravesical BCG treatment.
- Please explain in the manuscript why healthy subjects were not included for comparison in the data.
Thank you for the comment. There were several aspects we considered when collecting samples for the control group. First, we performed urethral catheterization, which is an invasive procedure with potential complications, such as urethral infection and pain. Unlike other sample collection methods, urethral catheterization is invasive and not suitable for healthy individuals without urological diseases, which could raise ethical concerns. Second, there is no appropriate public dataset containing catheterized urine samples from healthy individuals.
So, we chose urine samples collected from patients with BPH. Although the mid-stream urinary microbiome of BPH patients has been suggested to differ from that of normal controls [1], there is no evidence that the catheterized urinary microbiome of BPH patients is different from that of normal controls, unlike in other urological diseases such as neurogenic bladder dysfunction or interstitial cystitis [2]. Moreover, urethral catheterization is routinely performed in BPH patients in the operating room, making it easy to collect samples and obtain a sufficient number for appropriate analysis. BPH patients also primarily consist of elderly males, which matches the cohort of bladder cancer (BCa) patients; therefore, there was no significant age difference between the BPH and BCa cohorts (Table 1). Furthermore, to improve the quality of our control BPH group, we excluded samples from patients with urinary retention, infection, or high PSA levels, which may suggest possible prostate cancer, as these conditions can lead to changes in the urinary microbiome. These considerations are summarized and described in the limitations section of the Discussion.
--> [Page 10, line 42]: Added: “As we gathered samples from urethral catheterization, which is an invasive, possibly painful, and infection-inducing procedure, we were unable to collect samples from healthy controls without urological diseases. In addition, we couldn’t find any appropriate external dataset containing catheterized urine samples from healthy controls. Though it was not a perfect match for controls, BPH was our preferred choice, considering the ease of sample collection, similar age distribution, and male predominance in BCa. To improve the quality of the control group, we excluded patients in the control group with urinary retention, infection, and high prostate-specific antigen (PSA) levels, which suggest possible prostate cancer”.
- Discussion: Please include a more detailed explanation or possible molecular mechanisms elucidating the proposed relationship between the urinary microbiome, metabolic pathways (including the role of toluene degradation), and BCa.
Thank you for the comment. We made several modifications and additions on page 9 as follows:
- [Line 19]: Changed "relative" to "elevated".
- [Line 25]: Added: “Table S1” for referencing the data on hippuric acid.
- [Line 33] Added: “compared to controls”
- [Line 35]: Added: “In summary, we suggested and validated that an increase in toluene-degrading bacteria, such as Thauera or Desulfovibrio genera in the urinary microbiome, is related to a de-creased amount of toluene in urine, leading to a lower risk of bladder carcinogenesis. In addition,”.
[Minor points]
- Introduction – TNM– Please provide full title before using an abbreviation.
Thank you for the comment. We added an explanation for the abbreviation:
Page 2, line 6 - TNM stage → Tumor-node-metastasis (TNM) staging
- I did not see Table 1 in the manuscript, original images, or supplementary file(s).
Sorry for the inconvenience. We have checked and re-uploaded the tables, including Tables 1 and 2.
- Please remove the yellow highlights from pages 10, 11, and 15.
Thank you for the comment. We have corrected the issue.
- Yu, S.H. and S.I. Jung, The Potential Role of Urinary Microbiome in Benign Prostate Hyperplasia/Lower Urinary Tract Symptoms. Diagnostics (Basel), 2022. 12(8).
- Whiteside, S.A., et al., The microbiome of the urinary tract—a role beyond infection. Nature Reviews Urology, 2015. 12(2): p. 81-90.

Reviewer 2 Report
Comments and Suggestions for Authors
Min et al. Have provided an exceptional manuscript on how the urinary microbiome is influenced and influences the response to BCG treatment through mechanisms involving toluene degradation. The data is sound and the presentation of the results is good. There are however some issues that need to be addressed prior to publication.
Major comments
It is incomprehensible to why patients with BPH is used as a control group. There are several issues in using these patients as controls.
First, there is no compairison between these patients and a normal control group, although the patients are not malignant does not equal that they have a normal microbiome.
Second, if the authors want to use a group with pathology as controls, there should at least be similar treatment options for the groups so that there would be similar comparisons in what BCG treatment is influenced by/influenes the urinary microbiome.
The authors and the manuscript would therefor benefit greatly by comparing the urinary microbiome of the bladder cancer patients to healthy controls instead.
Although it would require extensive work, validating the influence of Bifobacterium species using a bladder cancer model in germ-free and gnotobiotic mice with and without BCG would strengthen the claims of the studies and provide a molecular framework of the study. This should at least be discussed at some point in the manuscript,
Comments on the Quality of English LanguageThere's just a couple of typos here and there
Author Response
Comments to reviewers
First of all, we would like to thank you for the detailed and thoughtful comments. We discussed the points raised by the reviewers and made some modifications and improvements. All points that we modified or added are highlighted in yellow.
[Reviewer 2]
- It is incomprehensible to why patients with BPH is used as a control group. There are several issues in using these patients as controls.
- First, there is no comparison between these patients and a normal control group, although the patients are not malignant does not equal that they have a normal microbiome.
- Second, if the authors want to use a group with pathology as controls, there should at least be similar treatment options for the groups so that there would be similar comparisons in what BCG treatment is influenced by/influences the urinary microbiome.
- The authors and the manuscript would therefor benefit greatly by comparing the urinary microbiome of the bladder cancer patients to healthy controls instead.
Thank you for the comment. There were several aspects we considered when collecting samples for the control group. First, we performed urethral catheterization, which is an invasive procedure with potential complications, such as urethral infection and pain. Unlike other sample collection methods, urethral catheterization is invasive and not suitable for healthy individuals without urological diseases, which could raise ethical concerns. Second, there is no appropriate public dataset containing catheterized urine samples from healthy individuals.
So, we chose urine samples collected from patients with BPH. Although the mid-stream urinary microbiome of BPH patients has been suggested to differ from that of normal controls [1], there is no evidence that the catheterized urinary microbiome of BPH patients is different from that of normal controls, unlike in other urological diseases such as neurogenic bladder dysfunction or interstitial cystitis [2]. Moreover, urethral catheterization is routinely performed in BPH patients in the operating room, making it easy to collect samples and obtain a sufficient number for appropriate analysis. BPH patients also primarily consist of elderly males, which matches the cohort of bladder cancer (BCa) patients; therefore, there was no significant age difference between the BPH and BCa cohorts (Table 1). Furthermore, to improve the quality of our control BPH group, we excluded samples from patients with urinary retention, infection, or high PSA levels, which may suggest possible prostate cancer, as these conditions can lead to changes in the urinary microbiome. These considerations are summarized and described in the limitations section of the Discussion.
--> [Page 10, line 42]: Added: “As we gathered samples from urethral catheterization, which is an invasive, possibly painful, and infection-inducing procedure, we were unable to collect samples from healthy controls without urological diseases. In addition, we couldn’t find any appropriate external dataset containing catheterized urine samples from healthy controls. Though it was not a perfect match for controls, BPH was our preferred choice, considering the ease of sample collection, similar age distribution, and male predominance in BCa. To improve the quality of the control group, we excluded patients in the control group with urinary retention, infection, and high prostate-specific antigen (PSA) levels, which suggest possible prostate cancer”.
- Although it would require extensive work, validating the influence of Bifobacterium species using a bladder cancer model in germ-free and gnotobiotic mice with and without BCG would strengthen the claims of the studies and provide a molecular framework of the study. This should at least be discussed at some point in the manuscript,
Our group is conducting another study based on a transcriptome cohort and has identified results similar to those of this study. We presented the preliminary findings at the 2024 AACR and have cited these results in the manuscript.
--> [Page 10, Line 12]: Added: “In another study from our group, we selected a bacterial strain from the Bifidobacterium genus and validated its antitumor efficacy through in vitro and in vivo experiments, including subcutaneous and orthotopic BCa mouse models. This study demonstrated elevated expression of pro-inflammatory genes and increased cell populations. Considering that BCG exerts its efficacy via pro-inflammatory genes and trained immunity in the urinary bladder, this study supports our findings.”.
- Yu, S.H. and S.I. Jung, The Potential Role of Urinary Microbiome in Benign Prostate Hyperplasia/Lower Urinary Tract Symptoms. Diagnostics (Basel), 2022. 12(8).
- Whiteside, S.A., et al., The microbiome of the urinary tract—a role beyond infection. Nature Reviews Urology, 2015. 12(2): p. 81-90.

Round 2
Reviewer 1 Report
Comments and Suggestions for Authors
The author has provided thorough responses and revisions to the issues raised. The paper is now ready for publication.